# Scaffold-Free Retinal Pigment Epithelium Microtissues Exhibit Increased Release of PEDF

**DOI:** 10.3390/ijms222111317

**Published:** 2021-10-20

**Authors:** Abdullah Al-Ani, Derek Toms, Saud Sunba, Kayla Giles, Yacine Touahri, Carol Schuurmans, Mark Ungrin

**Affiliations:** 1Department of Comparative Biology and Experimental Medicine, Faculty of Veterinary Medicine, University of Calgary, Calgary, AB T2N 1N4, Canada; aalani@ucalgary.ca (A.A.-A.); saud.sunba@ucalgary.ca (S.S.); kayla.giles678@gmail.com (K.G.); 2Alberta Children’s Hospital Research Institute, University of Calgary, Calgary, AB T2N 4N1, Canada; 3Alberta Diabetes Institute, University of Alberta, Edmonton, AB T6G 2E1, Canada; 4Biomedical Engineering Graduate Program, University of Calgary, Calgary, AB T2N 1N4, Canada; 5Cumming School of Medicine, University of Calgary, Calgary, AB T2N 4N1, Canada; 6Sunnybrook Research Institute, Toronto, ON M4N 3M5, Canada; yacine.touahri@sunnybrook.ca (Y.T.); carol.schuurmans@sunnybrook.ca (C.S.); 7Department of Biochemistry, University of Toronto, Toronto, ON M5S 1A8, Canada

**Keywords:** RPE, microtissue, retina, tissue engineering, ESC-RPE, ARPE-19, ophthalmology

## Abstract

The retinal pigmented epithelium (RPE) plays a critical role in photoreceptor survival and function. RPE deficits are implicated in a wide range of diseases that result in vision loss, including age-related macular degeneration (AMD) and Stargardt disease, affecting millions worldwide. Subretinal delivery of RPE cells is considered a promising avenue for treatment, and encouraging results from animal trials have supported recent progression into the clinic. However, the limited survival and engraftment of transplanted RPE cells delivered as a suspension continues to be a major challenge. While RPE delivery as epithelial sheets exhibits improved outcomes, this comes at the price of increased complexity at both the production and transplant stages. In order to combine the benefits of both approaches, we have developed size-controlled, scaffold-free RPE microtissues (RPE-µTs) that are suitable for scalable production and delivery via injection. RPE-µTs retain key RPE molecular markers, and interestingly, in comparison to conventional monolayer cultures, they show significant increases in the transcription and secretion of pigment-epithelium-derived factor (PEDF), which is a key trophic factor known to enhance the survival and function of photoreceptors. Furthermore, these microtissues readily spread in vitro on a substrate analogous to Bruch’s membrane, suggesting that RPE-µTs may collapse into a sheet upon transplantation. We anticipate that this approach may provide an alternative cell delivery system to improve the survival and integration of RPE transplants, while also retaining the benefits of low complexity in production and delivery.

## 1. Introduction

Retinal degenerative diseases are the most common cause of blindness in industrial countries, substantially impacting both quality of life and healthcare costs [1]. Indeed, in the US alone, age-related visual impairment has a financial burden of over $5.5 billion per year [2,3,4]. Age-related macular degeneration (AMD) is the leading cause of blindness in the developed world, affecting up to 20% of people over 65 years old [5,6,7,8]. Dry AMD is characterized by the irreversible loss of retinal pigment epithelium (RPE) cells, followed by a gradual loss of photoreceptors in later stages of the disease [1,9]. Unfortunately, there is currently no curative clinical treatment for AMD patients, and most current clinical interventions aim only slow progression of the disease [10].

The RPE monolayer provides a critical trophic niche to house and support photoreceptors. Strategically located between the choroid and visual retina, the RPE interacts with both endothelial and photoreceptor cells and orchestrates the transfer of nutrients and waste products in and out of the retina. Moreover, the RPE performs a wide range of additional and essential supportive functions, including recycling vital proteins and secreting key trophic factors [11,12,13,14,15,16]. In dry AMD, degeneration of the RPE results in progressive photoreceptor erosion and consequent vision loss. A promising therapeutic approach is to replace the diseased/degenerated RPE cells with healthy ones derived from stem cells [17,18]. Several animal trials established the ability of stem-cell-derived RPE cells to rescue the visual function of blind rodents with a dysfunctional RPE [19,20], inspiring subsequent human clinical trials [21,22,23].

In a groundbreaking study, Schwartz and colleagues transplanted human embryonic stem cell (hESC)-derived RPE cells subretinally into two patients with AMD and Stargardt macular dystrophy in a phase I/II clinical trial [22,24]. No signs of tumorigenicity, proliferation, or ectopic tissue formation were reported [24]. Two follow-up clinical trials were launched by the same group, which enrolled 18 patients and further demonstrated the safety of transplanting hESC-derived RPE subretinally into patients, with preliminary data suggesting some enhanced visual function [24,25]. 

While very encouraging, obstacles remain before RPE cellular therapy moves into routine clinical use. In particular, concerns have been raised regarding the survival, engraftment and function of RPE transplanted as a cellular suspension [6,23,25,26,27,28]. While modest vision improvement may still be possible despite low engraftment, this is far from a complete solution, especially as integration efficiencies may be further reduced when donor cells are transplanted into a degenerative “real-world” environment [29]. Furthermore, if only modest improvement is expected, any potential benefits must be weighed against the potential to damage or detach the delicate structure of the diseased retina during surgery. 

As one alternative, RPE transplanted as a tissue has been shown to be superior in terms of morphology, physiology and survival [6,26,27,30]. However, transplantation of RPE tissues represents a greater surgical and technical challenge [26,31,32,33,34]. For instance, concerns have been raised regarding the use of scaffolds to support these delicate structures during delivery, as they have been associated with inflammation and RPE detachments from the choroid [26]. Scaffold-free stem cell derived RPE sheets have been generated by several groups to address the limitations associated with transplanting cell suspension and scaffolds; following several encouraging animal trials [26], Mandai and colleagues were able to derive and successfully transplant an RPE sheet from patient iPSCs [35]. Although scaffold-free RPE sheets show therapeutic potential, they are technically challenging and labor-intensive to produce, and their transplantation is surgically invasive as they are too large to be subretinally injected [26,32]. Following the approach of RPE sheets will thus impose economic and infrastructure-related barriers to widespread accessibility. We thus propose that combining the simplicity of cell suspension with the superior performance of engineered sheets, will yield a process that is cost-effective, widely accessible and easily scalable. 

Here, we lay the foundations for an alternative approach to RPE transplantation, using engineered scaffold-free RPE microtissues (RPE-µTs) with dimensions suited for delivery via the ultra-fine needles used in subretinal injections. Importantly, our RPE-µT expresses key RPE molecular markers and deposits physiologically relevant ECM molecules.

## 2. Results

In this study, we utilized two RPE cell sources: ARPE-19 and human embryonic stem cell (hESC) derived RPE cells. ARPE-19 is a spontaneously arising human cell line that is widely used in the field as it recapitulates key aspects of human RPE behavior and function in vitro and in vivo [36,37]. RPE derived from hESC is a new model that is more complex to generate, but it better mimics endogenous RPE behavior and has significant clinical potential [38,39]. 

### 2.1. RPE-µTs Form Efficiently and Do Not Require Adhesion to a Culture Surface

Size-controlled 3D RPE-µTs were formed in our centrifugal-forced aggregation platform (Figure 1a) [40]. Based on free-energy minimization models [41,42] and consistent with our previous observations of microtissue formation that were obtained using other cell types [43], RPE cells were dispensed into microwells and allowed to coalesce for 8 days (Figure 1b,c). Using the established ARPE-19 cell line, microtissues formed after only 4–6 days, whereas hESC-derived RPE required more time (6–8 days) to form coherent RPE-µT. To ensure consistency across experiments, we consequently cultured ESC-derived RPE and APRE-19 cells in microtissue and adherent cultures for 8 days. We were able to tightly control the diameter of the formed RPE-µTs by adjusting the seeding density of the cells (Figure 1d,e). As expected, microtissue volume correlated well with input cell numbers (R^2^ = 0.98) (Figure 1f).

### 2.2. RPE-µT Size Influences mRNA Levels of Key RPE Functional Genes

After confirming that RPE-µTs of various sizes could be produced, we asked which size had the potential of superior function and transplantation potential based on gene expression profiles. We used RT-qPCR to measure expression levels of a panel of 20 key RPE marker genes in RPE-µTs formed from 25 (R25), 100 (R100) and 400 (R400) cells. Strikingly, R100 and R400 RPE-µTs expressed higher levels of key genes that affect photoreceptor stability (*FGF2, PEDF* and *BDNF*), choroid stability (*TGFB, TIMP3* and *VEGF*) and general RPE functions (*CCL2, CFH, IL8, LHX2, MYRIP* and *LOXL*) compared to R25 (Figure 2a). However, the difference between R100 and R400 RPE-µTs in terms of mRNA levels of the investigated genes was not statistically significant for most genes, except for *CFH*, where R100 had higher mRNA levels.

### 2.3. R100 Demonstrates Superior Survival after Passage through a 30-Gauge Needle and Reestablishes a Monolayer

Having identified a broad gene expression optimum in the R100–R400 range, we then assessed the feasibility of delivering our 3D RPE-µTs via injection. R100 and R400 survival was assessed following the passage through a 30-gauge needle (159 µm inner diameter) at a rate of 50 µL/minute (a rate commonly used for rodent subretinal injections [44,45]) (Figure 2b,c). To measure live:dead cells, RPE-µTs were stained with DAPI to label nuclei; fluorescein diacetate (FDA) to label live cells; and propidium iodide (PI), which is taken up by dead cells (Figure 2d). Injected R100 exhibited over an order of magnitude greater live:dead staining ratio than R400 (Figure 2e). Control samples for R100 could not be calculated as most pixels in the FDA channel were saturated and circumstances beyond our control prevented us from repeating this series of experiments. However, we observed high viability in all RPE-µT and feel this omission does not change our interpretation of the data. Based on these results, R100 RPE-µTs were used for all subsequent experiments.

In vivo, RPE functions as a columnar epithelium that is one cell in thickness. We hypothesized that given the opportunity (for example, via seeding into an RPE niche left unoccupied due to loss of the endogenous RPE), our RPE-µTs would self-organize to re-constitute the natural tissue configuration. The composition of Matrigel, although not a perfect substitute for Bruch’s membrane, is largely comparable: both are primarily composed of laminin, collagen type IV and fibronectin [46,47]. It is unlikely that small differences in composition would affect binding and adhesion of RPE to Matrigel since the primary mechanism of adhesion requires laminin to bind to RPE’s integrin [48]. We therefore plated RPE-µTs on coverslips coated with Matrigel. Time-lapse confocal microscopy revealed RPE cells migrating out of the microtissues within 24 h of their deposition and aggregate height decreasing by approximately half. The most dramatic effect was observed between 24 and 48 h (Figure 2f). Integrated pixel density significantly increased when comparing images of RPE-µTs at 0 and 48 h, further validating that the RPE cells are occupying more space in the field of view, likely due to spreading from the initial aggregate (Figure 2g).

### 2.4. RPE-µTs Express Key RPE Markers

To extend molecular characterization beyond the transcriptional level, we immuno-stained ARPE-19 RPE-µTs for key RPE functional markers. Cellular retinaldehyde-binding protein (CRALBP—Figure 3a) and lecithin retinol acyltransferase (LRAT—Figure 3b) were both detected within microtissues, two key proteins that play a vital role in the visual recycling function of RPE [11,49,50,51]. Bestrophin-1 (Best1) was also detected (Figure 3d), typically found in the basolateral side of the RPE monolayer where it plays a role in calcium ion regulation [52,53]. The presence of zonula occludens-1 (ZO-1) in the classical “chicken-wire” pattern (Figure 3e) suggests the formation of tight junctions within RPE-µTs [11,54]. Moreover, we were able to detect the expression of melanocyte inducing transcription factor (MITF—Figure 3c), an RPE specific transcription factor that is crucial for RPE development and function [55]. The extracellular matrix components laminin, collagen IV and fibronectin (Figure 3f–h) are known to be secreted by RPE and to contribute to both Bruch’s membrane on the basal side and the interphotoreceptor matrix on the apical side [56,57]. However, retinal-pigment-epithelium-specific 65 kDa protein (RPE65) was not detected in our ARPE-19 RPE-µT. This finding is consistent with literature reports that the ARPE-19 cell line expresses low and sometimes undetectable levels of RPE65 [37].

### 2.5. RPE-µTs Upregulate Transcription of Desirable Secreted Factors

We next sought to characterize RPE-µTs’ signaling potential in more depth by comparisons with conventional monolayer cultures. A group of 21 RPE-specific genes was selected to evaluate the function of our microtissue, and the genes were categorized into secreted factors for choroid stability (*FASL, TIMP3, TGFB, VEGFA* and *PDGF-AA*), secreted factors for photoreceptor stability (*PEDF, IGF1, BDNF, GAS6* and *FGF2*) and general RPE functional markers (*RPE65, CFH, TRMP1, BEST1, FGF2R, MYRIP, CCL2, IL8, LHX2, LOXL* and *KDR*) [58] (Figure 4a). When comparing ARPE-19 microtissue to adherent culture, RT-qPCR data indicated upregulation of mRNA levels of *PEDF* and *IGF-1* and downregulation of *VEGF* and *TGF-β* (Figure 4a). Interestingly, RT-qPCR data also indicated the upregulation of *RPE65* transcript, and downregulation of *FGF-2, GAS6, PDGF-AA, CCL2, IL-8, LHX2, LOXL* and *KDR* transcripts in RPE-µTs compared with 2D controls.

We next collected conditioned media (CM) to assess secreted protein levels from RPE monolayer cultures versus microtissues, using Multiplexing Laser Bead Technology. Strikingly, the ARPE-19 microtissue secreted significantly more PEDF and IGF-1 proteins and less VEGF than the standard adherent culture (Figure 4b). 

While the ARPE-19 cell line is widely used to recapitulate RPE behavior and many of its key functions [36,37], significant limitations have been identified including low expression of important functional genes such as *PEDF, RPE65* and *VEGF*, as well as a lack of pigmentation [36,37,38,39,59]. We correspondingly sought to validate our findings using more therapeutically relevant RPE cells, such as hESC-derived RPE [6,22,23,26,27]. We thus derived RPE from hESCs as per the protocol published by Maruotti et al., with some modifications (see Materials and Methods for details) [60]. We validated the successful derivation of RPE cells, using RT-qPCR analysis of key RPE genes (Figure 5a), such as *BEST1, RPE65, VEGF, PEDF, LHX2, CCL2* and *DUPS4,* as well as immunostaining and examining their morphology and pigmentation. As expected, our differentiated RPE cells showed dramatic increases in expression for these genes and a downregulation of the pluripotency gene *OCT4* (Figure 5b). In addition to adopting a hexagonal morphology, our differentiated RPE had increased pigmentation [58], as RPE-µT compared to monolayer cultures (Figure 5c). Our differentiated RPE cells stained positively and specifically for key RPE markers, including RPE65, LRAT, CRALBP, Melanopsin, Sox9 and Best1, as well as ZO-1, thus suggesting the formation of tight junction (Figure 5d). 

Just as with the ARPE-19 cell line, R100 RPE-µTs were produced from these hESC-derived RPE cells and they were compared to adherent culture in terms of mRNA levels and secreted proteins for key RPE functional markers. We observed consistent results at both the transcript and protein levels between ARPE-19 and hESC-derived RPE cells. When comparing the RT-qPCR data from Figure 4 and Figure 5, we observed consistent trends by RPE-µTs produced from both cell types; namely, 17 out of the 21 investigated genes demonstrated the same mRNA expression trend. Moreover, hESC-derived RPE microtissues also upregulated *PEDF* and *IGF-1,* and downregulated *VEGF* mRNA levels (Figure 5e). Additionally, hESC-derived RPE microtissue also appeared to upregulate mRNA levels for *RPE65* and *MYRIP,* while downregulating *TGF-β, BDNF, FGF2, GAS6, KDR, LOXL, CFH, IL-8* and *LHX2* (Figure 5e). 

Similar to the ARPE-19 microtissue, cells in the hESC-derived RPE-µTs secrete 3X more PEDF (*p* < 0.01) and less than 1% of VEGF (*p* < 0.01) than standard adherent culture (Figure 5f). Based on these secreted factor profiles, we predicted that CM from RPE-µT cultures would increase photoreceptor survival, using a dissociated mouse retina assay [61]. After 48 hours, photoreceptor-enriched in vitro cultures demonstrated superior survival with RPE-µT CM compared to CM collected from monolayer cultures (1.41 ± 0.11-fold increase, *p* < 0.05) and this effect was exaggerated by 72 h with a nearly two-fold increase in cell survival, using RPE-µT CM (*p* < 0.01; Figure 5g).

## 3. Discussion

In this work, we were able to produce size-controlled and scaffold-free RPE-µTs efficiently and reproducibly from both the patient-derived ARPE-19 cell line, and RPEs derived de novo from hESCs. While multiple groups have demonstrated the utility of RPE transplantation in animal and human trials, transplanting a single-cell suspension exhibits relatively poor survival, engraftment and function [6,19,20,21,25,26]. This outcome is not surprising as enzymatically-dissociated cells are stripped of their ECM and intercellular connections with neighboring cells [62], potentially reducing both viability and function [28,63]. Transplant of engineered RPE sheets has shown promise in recent clinical trials, with encouraging initial reports demonstrating high survival and integration [6,27,35], reinforcing the importance of intercellular connections and endogenous ECM. However, this approach is comparatively expensive, surgically invasive, technically challenging and time consuming in comparison to traditional subretinal injections [26], which raises concerns around accessibility, scalability and economic viability [64]. We hypothesized that a practical solution could be achieved by using engineered injectable RPE-µTs that combine the simplicity of production and handling of single cell suspensions, with the intercellular connections and endogenous ECM production of RPE sheets. 

The production of RPE-µT is efficient and is linearly scalable with microwell surface area [43]. For instance, the 24-well plate format of the microwell system can produce up to 28,800 RPE-µTs per plate. Keeping in mind that Schwartz and colleagues injected between 50,000 and 150,000 RPE cells into each eye in their clinical trial [23], a 24-well plate format of the microwell system can theoretically produce sufficient RPE-µTs for 19–57 eyes. However, future work is needed to assess the function, stability and storage of large batches of RPE-µTs that would be necessary to develop scalable clinical therapies. 

R100 RPE-µTs exhibited both a near-optimal transcriptional profile and the ability to survive delivery through a 30-gauge (159 µm inner diameter) needle (Figure 2), which is commonly used in subretinal transplantation in mice [44,45]. While larger bore needles may be utilized in larger organisms [22], there are advantages to maintaining a consistent delivery approach across model organisms and into the clinic, and in the absence of enhanced transcriptional profiles, bigger RPE-µTs are not necessarily better as they may experience limited distribution in the subretinal space [65]. We suspect the increased cell death in larger aggregates is due to their proximity to the needle walls where fluid shear stress is maximal [66].

The upregulation of *PEDF* and *IGF-1* we identified in RPE-µTs has important therapeutic implications, as both molecules provide potent photoreceptor pro-survival signals [67,68,69,70,71,72,73,74,75] and have been shown to rescue photoreceptors in retinal degeneration animal models [12,70,74]. Enriched photoreceptors have been reported to rapidly degenerate in vitro, partially due to the lack of support from RPE [76,77]. While our survival assay largely simplifies the degenerative retinal environment where RPE loss may also be occurring, our results from this experiment suggest that the trophic factors measured in CM are indeed biologically active. The secretome of our RPE-µTs thus appears to provide greater support for photoreceptors when compared to RPE cells grown as a monolayer. We hypothesize that PEDF and IGF-1 are likely at least partially responsible, although this prediction would require further experimentation in a more sophisticated model of retinal degeneration. Increased secretion of PEDF in RPE-µTs is of particular interest, as this neurotrophic factor has been shown to have clear therapeutic potential in the treatment of retinal degeneration by preventing photoreceptor apoptosis [67,68,78,79]. 

Similarly, the downregulation of *VEGF* and *TGF-*β also represent desirable phenotypes, as overproduction of both angiogenic factors is linked to choroidal neovascularization, wet AMD and photoreceptor death [9,80,81,82,83,84]. In addition to the downregulation of *TGF-*β and VEGF, our hESC-derived RPE-µTs downregulated the transcription of *IL-8* and *CFH*, two genes expressed by RPE to modulate the immune response [11,85]. Interestingly, the upregulation of both genes has been linked to AMD [86,87,88]. Decreased expression of *LOXL* and *LHX2* in RPE-µTs suggests that the cells in our microtissues are healthy and mature [89,90,91].

The findings of the RPE-µT spreading experiments (Figure 2f) are consistent with a study by Beaune and colleagues that revealed cancer cell aggregates are able to spread on a two-dimensional surface [92]. We are not aware of any previous studies that analyzed the spreading of RPE aggregates on a 2D structure. This finding is important as it demonstrates the potential of using RPE-µTs in future clinical trials as a potential therapy for disorders such as age-related macular degeneration. In comparison to Bruch’s membrane, Matrigel lacks collagen type V and perhaps more importantly, may significantly differ from aged or diseased Bruch’s membrane, especially in patients with AMD. However, Matrigel remains one of the most common in vitro substances used to mimic authentic basement membranes [93] and given the robust production of Bruch’s membrane components by RPE-µTs (Figure 3), it seems likely that ECM deposition and remodeling will also take place as aggregates reorganize into a monolayer within the retina.

These data presented here support the potential for efficient delivery of RPE in the form of injectable RPE-µT, which would then re-organize to colonize RPE niches left unoccupied by injury and disease. Further testing in animal models of retinal degeneration will be necessary to confirm the potential of our RPE microtissues, to characterize to what extent injected RPE-µTs re-establish a monolayer and support the retina after transplantation. Based on the gene expression changes we observed (Figure 4 and Figure 5) in RPE-µTs relative to monolayer-on-plastic cultures, we anticipate that in the context of endogenous RPE degeneration, transplantation of our RPE-µTs will enhance photoreceptor protection over that yielded by delivery of RPE as a single-cell suspension, while remaining significantly less technically challenging than delivery of RPE in sheet form. Importantly, R100 RPE-µT size (diameter approximately 57 ± 7.3 µm) is significantly less than the range over which the focal length of the eye adjusts during normal accommodation [94], and represents only a small fraction of the lens-retina distance of 1.7 cm. While the reorganization of RPE from RPE-µT to monolayer we observed is highly promising for their ability to re-occupy vacant RPE niches, even a complete failure of RPE-µT to remodel into a natural epithelium would still result in net benefits to the recipient if they are able to rescue loss of endogenous photoreceptors via secreted signals.

## 4. Materials and Methods

### 4.1. Adherent and 3D Microtissue RPE Cultures

ARPE-19 cells were seeded in a 24-well plate (VWR, Mississauga, ON, Canada. cat #82050-892) at 120,000 cells/well (60,000 cells/cm^2^), reached confluency at ~120,000 cells/cm^2^, and cultured for 8 days in 1 mL of ARPE-19 culture media that consisted of: DMEM/F-12 with HEPES (Gibco, Mississauga, ON, Canada. cat # 11330057); 10% FBS VWR, Mississauga, ON, Canada. cat # 97068-085; and 1% penicillin/streptomycin (Gibco, Mississauga, ON, Canada. cat # 15140). Media change was performed every 48 hours replacing the old media with 1 mL of fresh media.

The hESC-derived RPE cells were differentiated from hESCs, as previously described by Maruotti and colleagues [60], with the following modifications: the HES-2 cell line [95] was grown to confluence under 5% CO_2_ and 5% O_2_ in mTeSR1; during induced differentiation; and a concentration of 50 nM chetomin (CTM) was used. Cells were grown to a low passage number (3 or 5) before being cryogenically stored. For all experiments, hESC-derived RPE cells were cultured in an identical manner to ARPE-19 cells, using RPE media that consisted of 70% DMEM (Gibco, Mississauga, ON, Canada. cat # 11965), 30% F12 (Gibco, Mississauga, ON, Canada. cat # 11765), 2% B-27 supplement (Gibco, Mississauga, ON, Canada. Cat # 17504) and 1% antibiotic (Gibco, Mississauga, ON, Canada. cat # 15140).

To estimate oxygen delivered to cultured cells under the aforementioned experimental conditions, we utilized our previously published method [58,96]. For the purpose of the calculation, we used 42 amol∙cell^−1^∙s^−1^ as the RPE oxygen consumption rate [97]. These calculations yielded that RPE cultured under the outlined conditions receive sufficient amount of oxygen, with a local oxygen concentration of 7.03 × 10^−5^ mol/L at the cells and a maximum oxygen delivery rate of 75.5 amol∙cell^−1^∙s^−1^. 

To generate RPE microtissues, adherent RPE cells were dissociated by adding 1 mL of warm TrypLE Express Enzyme (Gibco, Mississauga, ON, Canada. cat # 12604013) per well and incubating for 5–10 min at 37 °C. This single cell suspension was seeded into 24-well AggreWell plates (STEMCELL Technologies, Vancouver, BC, Canada. cat # 34411), as previously described [40,43]. Cells were cultured in 1 mL of medium at 37 °C for 8 days. The medium was completely changed every two days. Microtissue size was controlled by modulating the number of cells seeded per microwell, and is reflected in the “Rxx” designation (e.g., R25 RPE-µTs are formed by seeding 25 cells per microwell, R400 represents RPE-µTs formed from 400 cells apiece, etc.). Total cell number per well was calculated by multiplying the desired number of cells per microtissue by the number of microwells per well (1200).

After allowing 8 days for the cells to consolidate, we washed the RPE-µTs out of the wells, using a wide-bore 1000 µL pipette and repeated pipetting up and down. The resuspended RPE-µTs were transferred into a 1.5 mL Eppendorf tube to allow the RPE-µTs to gravity settle. The resulting RPE-µTs were then dispersed over a glass slide and imaged to measure their size, using a 10X objective on an Olympus CKX41 microscope. RPE-µT size was assessed from calibrated photomicrographs using ImageJ software [40,43]. Finally, the volume of produced RPE-µTs was modeled by using the measured diameter and correlating it to the number of RPE cells seeded.

### 4.2. RNA Extraction and cDNA Synthesis

To harvest the RPE-µTs and adherent cultures, cells were dissociated by adding 0.4 mL of warm TrypLE Express Enzyme (Gibco, Mississauga, ON, Canada. cat # 12604013) to each well and incubating for 5 minutes at 37 °C. Recovered cells were then centrifuged at 200× *g* and the resultant cellular pellets were then stored at −80 °C to be used for mRNA extraction. The mRNA was isolated by using a Total RNA Purification Kit (Norgen Biotek, Thorold, ON, Canada. cat # 37500) and quantified on a NanoDrop 2000 spectrophotometer (Thermo Fisher Scientific). The isolated RNA was reverse-transcribed by using iScript Reverse Transcription Supermix for RT-qPCR (Bio-Rad, Mississauga, ON, Canada. cat # 1708840). The resultant cDNA was then used to carry out real-time quantitative reverse transcription PCR (RT-qPCR) to compare gene-expression profiles between an RPE-µT and adherent culture (Section 2.4 and Section 2.5). SYBR Green RT-qPCR was carried out with technical triplicates using 7500 Fast Real-Time PCR System (see Reference [58] for primer sequences), and analyzed with the 2^−ΔΔCT^ method [98], and with stable internal reference gene (CNTF).

### 4.3. Immunohistochemistry

RPE-µTs were fixed by incubating in 4% paraformaldehyde (PFA) for 10 min at room temperatures. RPE-µTs were then washed in PBS for 5 min at room temperature three times. RPE-µTs were then incubated in 0.5% Triton X-100 detergent for 5 min at room temperature, followed by three washes with PBS. Primary antibodies were then added to the RPE-µTs at the proper concentration in PBT (0.1% TritonX-100 in PBS) and 1% BSA solution (primary antibody details including concentration and cat # can be found in Appendix A). The microtissues were then incubated in the primary antibody solution overnight at 4 °C, after which they were washed three times in PBS, followed by blocking in 1% BSA for 10 min at room temperature. The RPE-µTs were then incubated in the secondary antibody solution (secondary antibody and 1% BSA in PBS) for one hour at room temperature, before again being washed three times with PBS. The nuclei of the RPE-µTs were stained, using DAPI stain, before being mounted on slides for imaging. Stained RPE-µTs were imaged by using a Leica SP8 spectral confocal microscope.

### 4.4. Microtissue Injection Modeling

To verify the feasibility and practicality of RPE-µT delivery via subretinal injection, we assessed survival after delivery through the bore of a 30-gauge needle. Microtissue of various sizes (R100 and R400) were harvested from the AggreWell plate 8 days post-seeding and approximately 200 microtissue were resuspended in 10 µL of PBS. The microtissue in PBS were loaded to 100 µL syringe and injected manually through a 30-guage needle (BD, Holdrege, NE, USA. cate # 305128) at a rate of 10 µL/minute. RPE-µT survival was evaluated by using the live/dead cell assay described below (Figure 2b,c). 

Image quantification for the live/dead assay employed a custom ImageJ macro (see Appendix A for code and representative example images; analysis was carried out with default settings unless otherwise specified). Signal in the Hoechst (nuclear staining) channel was blurred prior to thresholding, followed by hole-filling and dilation–erosion cycles to give a smooth outline capturing the entire structure. Watershed segmentation was then employed to separate structures in contact with one another, followed by particle analysis that calculated a mean signal intensity for each particle in the green (live cell) and red (dead cell) channels.

### 4.5. Secreted Protein Quantification

CM was collected from generated RPE-µTs and standard RPE adherent cultures during standard media change (48 h after introducing the media). The CM was then stored at −80 °C until analysis. Multiplexing Laser Bead Technology (Eve Technologies) was performed on CM to estimate the concentration of proteins present.

### 4.6. Viability Assay

Live/dead staining solution contained: Hoechst (Thermo, Mississauga, ON, Canada. cat # 62249) as a nuclear stain, fluorescein diacetate (FDA) (Thermo, Mississauga, ON, Canada. cat # F1303) to indicate live cells, and propidium iodide (PI) (Thermo, Mississauga, ON, Canada. cat # P3566) to indicate dead cells. All three regents were combined in a staining solution (1/50 of each stain in PBS). Cells and RPE-µTs were incubated in 50 µL of staining solution for 5 min at room temperature. Cells or microtissue were washed three times with PBS and imaged by using an Olympus IX83 Microscope at 200× magnification, using MicroManager software [99]. 

Photoreceptors were isolated, enriched and cultured as previously described [61]. Briefly, mouse retinas were dissected from the eyes of postnatal day 4 CD1 mice obtained through a secondary-use protocol approved by the Animal Care Committee at the University of Calgary according to IACUCC standards. Retinas were isolated from multiple pups in one litter, polled together and enzymatically dissociated. Photoreceptors were magnetically enriched by using 3 µg/mL of PE-conjugated anti-CD73 antibody (BD cat. # 550741) on an EasySep cell-separation system (StemCell Technologies cat. # 18554). Photoreceptors were cultured in 96-well plates coated with poly-d-lysine at 3.1 × 10^5^ cells/cm^2^. Images for photoreceptor viability assay were processed by using a Cell Profiler pipeline that quantified the nuclei that are co-localized with either live or dead stains to determine the percentage of live cells [100] and results were confirmed manually by an evaluator who was blinded to the experimental conditions.

### 4.7. Attachment Assay

To observe how RPE-µTs interacted with a matrix resembling Bruch’s membrane, we imaged eight-day-old APRE-19 RPE-µTs after seeding on Matrigel-coated glass coverslips. A stable ARPE-19 cell line expressing nuclear mCherry was generated and used to form RPE-µTs. Glass coverslips were coated with Matrigel by placing them individually in a 24-well plate followed by 2 mL of a solution of growth factor-reduced Matrigel (Corning, Bedford, MA, USA. cat #354230) diluted 1:100 in DMEM-F12 and incubated at 37 °C for 1 h. Coating solution was removed and 2 mL RPE media was added to each well. After 8 days, RPE-µTs were gently removed from microwells with a P1000 pipette and placed in the wells containing fresh media. RPE-µTs were imaged by using Leica SP8 spectral confocal microscopy.

### 4.8. Statistical Analysis

Data were reported as mean ± SD. Data in this study were analyzed with GraphPad Prism (v.7, San Diego, CA, USA) and R statistical software (v.3.5.1, Vienna, Austria [101]). Calculations of Ct values were carried out by using Microsoft Excel (v. 16.53, Redmond, WA, USA). Information about the specific statistical methods can be found in the text and figure legends. Results presented as “fold expression” comparing RPE-µTs and monolayers were analyzed with a Wilcoxon matched-pairs signed-ranks test to determine whether the mean ∆Ct differed (Appendix A). For this study, *p* < 0.05 was considered statistically significant.

## Figures and Tables

**Figure 1 ijms-22-11317-f001:**
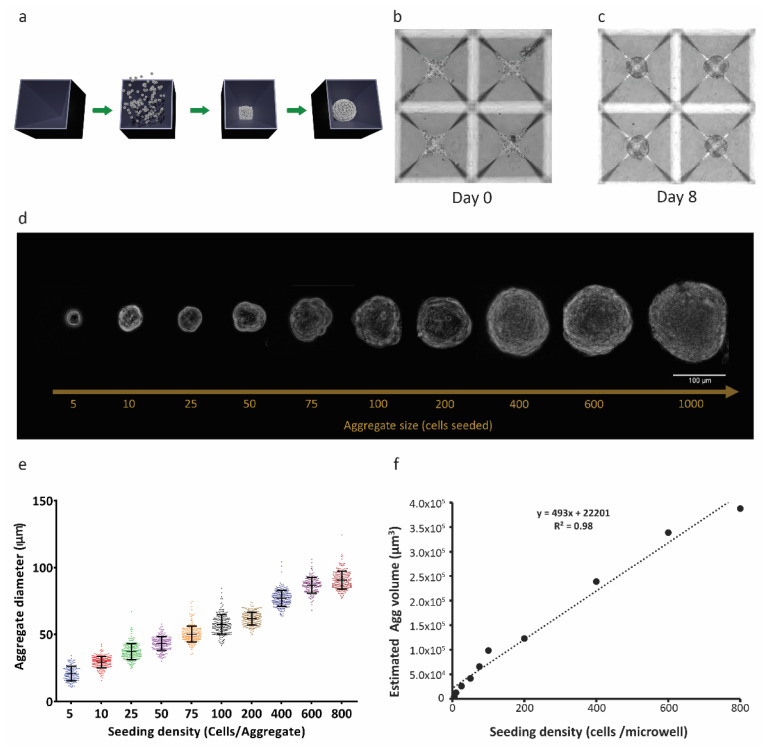
Three-dimensional RPE-µTs show consistent size control. (**a**) ARPE-19 cell suspensions were centrifuged into 400 µm microwells and cultured to produce RPE-µT. (**b**) Cells settled at the bottom of the microwells and (**c**) formed robust spherical RPE-µTs by 8 days in culture. The size of the 3D RPE-µTs correlated with the number of cells seeded per microwell, as evidenced either visually (**d**) or via quantitative analysis (**e**). Each group of seeded cells per RPE-µT was significantly different from all others based on a one-way ANOVA followed by Tukey’s post hoc test (n = 100, *p* < 0.001). (**f**) Approximating RPE-µTs as spheres with the observed diameters, we observed that their calculated volume showed a strong linear correlation with the number of cells from which they were formed (R^2^ = 0.98).

**Figure 2 ijms-22-11317-f002:**
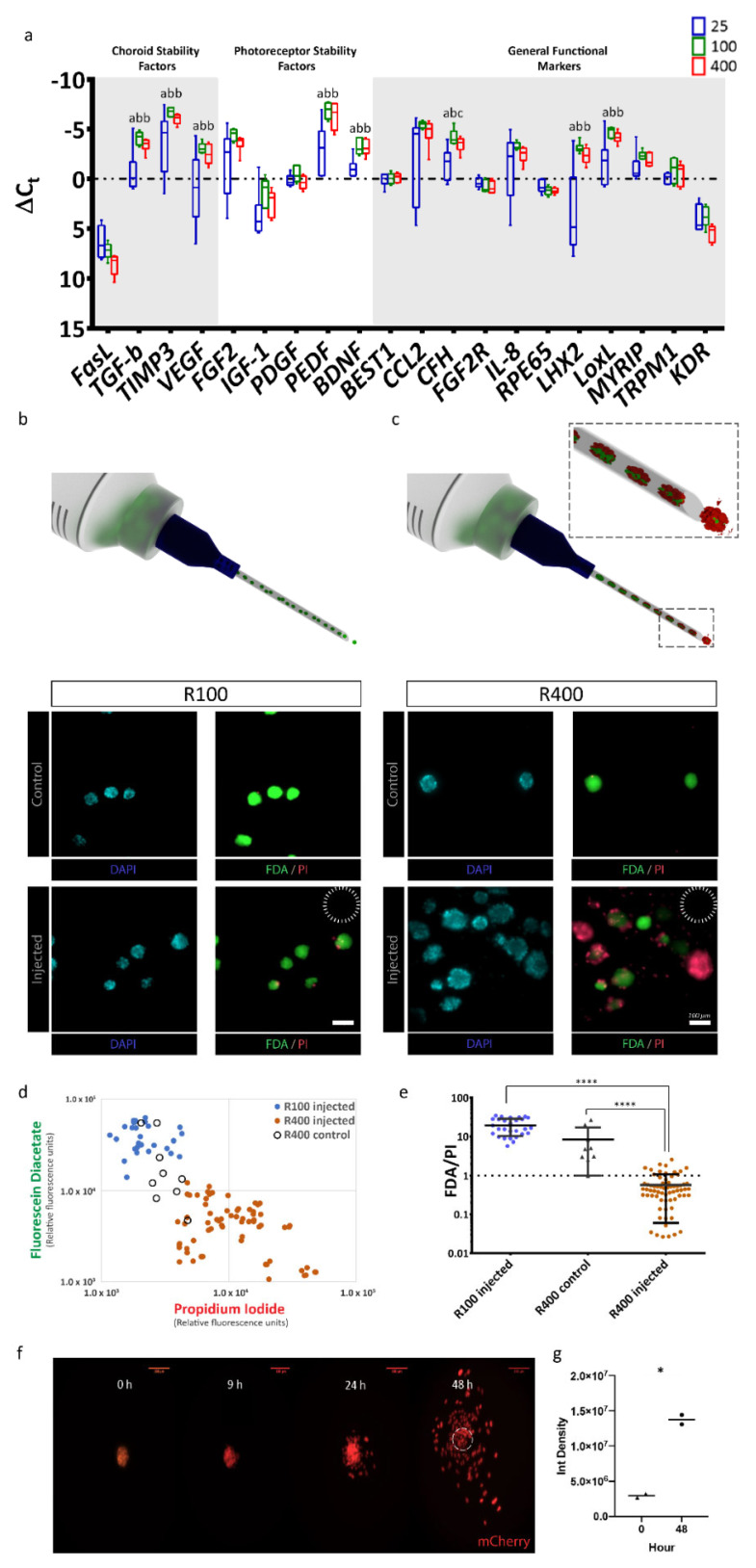
R100 RPE-µTs exhibit the desired gene-expression profile and tolerance for delivery through a 30-gauge needle. (**a**) RT-qPCR was conducted to compare the mRNA expression profiles of R25, R100 and R400 RPE-µTs (n ≥ 4) for genes previously reported as photoreceptor stability factors, choroid stability factors and general functional markers. Results are presented as ΔCt values to facilitate direct comparisons among R25, R100 and R400 RPE-µT sizes, shown in blue, green and red, respectively. Kruskal–Wallis ANOVA followed by Dunn’s test for multiple comparisons was conducted to compare ΔCt values within each gene (* *p* < 0.05 for groups that were statistically significant). Statistically distinct groups were labeled with different letters (i.e., “a” and “b” are statistically different). In order to assess potential for minimally invasive delivery via subretinal injection, (**b**) R100 and (**c**) R400 RPE-µTs were injected through a 30-gauge needle (inner diameter of 159 µm, dotted circle) at 50 µL/minute. Subsequent staining for fluorescein diacetate (FDA-green) and propidium iodide (PI-red) cells revealed that R100 survive the injection while R400 showed inferior survival. Quantification of FDA and PI further demonstrated the superiority of R100 RPE-µT. (**d)** Scatter plot of log FDA vs. log PI with each data point representing injected R100 (blue dot; n = 28 aggregates), control R400 (empty black circles; n = 9 aggregates) and injected R400 (orange dots; n = 68 aggregates). The mean fluorescence for the green and red channel of each RPE-µT was quantified by using a custom-made Image J plugin that automated fluorescence measurement for each aggregate (details available in the materials and methods section). (**e**) The ratio of FDA (green) to PI (red) fluorescence for each measured aggregate was calculated and Mann–Whitney U test was conducted to evaluate the statistical significance (**** *p* < 0.0001; error bars represent standard deviation). R100 control samples could not be quantified due to a high percentage of saturated pixels in the FDA channel (see text for more details). (**f**) Engineered RPE-µTs expressing nuclear mCherry spreading on Matrigel-covered slip imaged over a 48-hour period. (**g**) Integrated density of multiple pictures (n = 7) for both time points were calculated. Statistical significance determined by Mann–Whitney U test.

**Figure 3 ijms-22-11317-f003:**
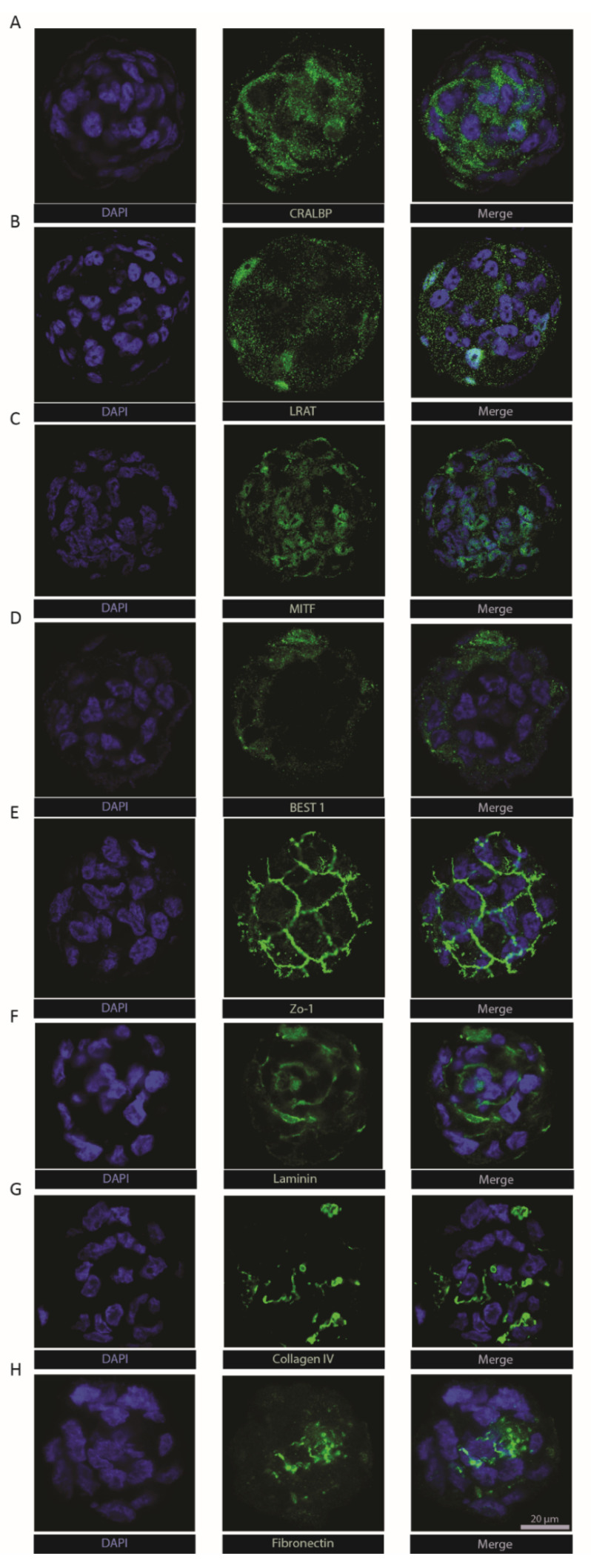
Engineered RPE-µTs maintain the expression of hallmark identity and functional RPE markers. Immunostained ARPE-19 RPE-µTs with antibodies against (**A**) CRALBP, (**B**) LRAT, (**C**) MITF, (**D**) BEST1, (**E**) ZO-1, (**F**) laminin, (**G**) collagen IV and (**H**) fibronectin. All samples were counterstained with DAPI and are shown at the same magnification; scale bar = 20 µm.

**Figure 4 ijms-22-11317-f004:**
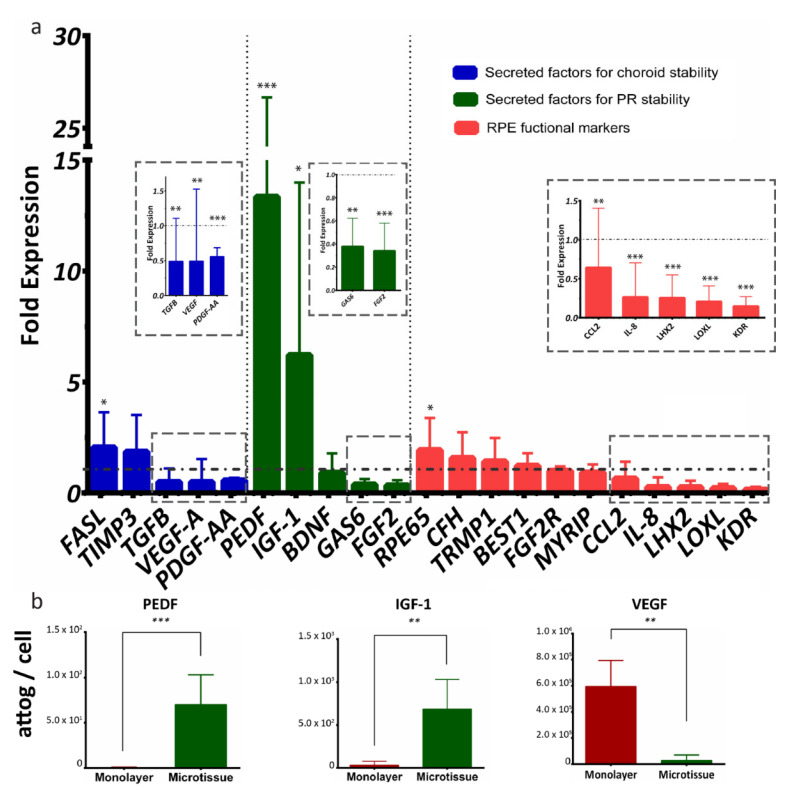
ARPE-19 RPE-µTs upregulate the expression of desirable photoreceptor trophic factors. (**a**) RT-qPCR was conducted to compare the expression profile of key genes between our 3D RPE-µTs and standard adherent culture of ARPE-19 (n = 8–13). The results are presented as “fold expression” values to showcase the extent of expression enhancement of certain genes in our RPE-µTs (bar graphs) compared to adherent culture (dotted line), with gene expression normalized to an endogenous reference gene (CNTF). A Wilcoxon matched-pairs signed-ranks test was used to compare ΔCt values between RPE-µTs and adherent culture for each gene. (**b**) Conditioned media was analyzed for levels of secreted PEDF, IGF-1 and VEGF and statistical significance determined by using a Mann–Whitney U test (n = 5); * *p* < 0.05, ** *p* < 0.01, *** *p* < 0.001. Error bars represent standard deviation.

**Figure 5 ijms-22-11317-f005:**
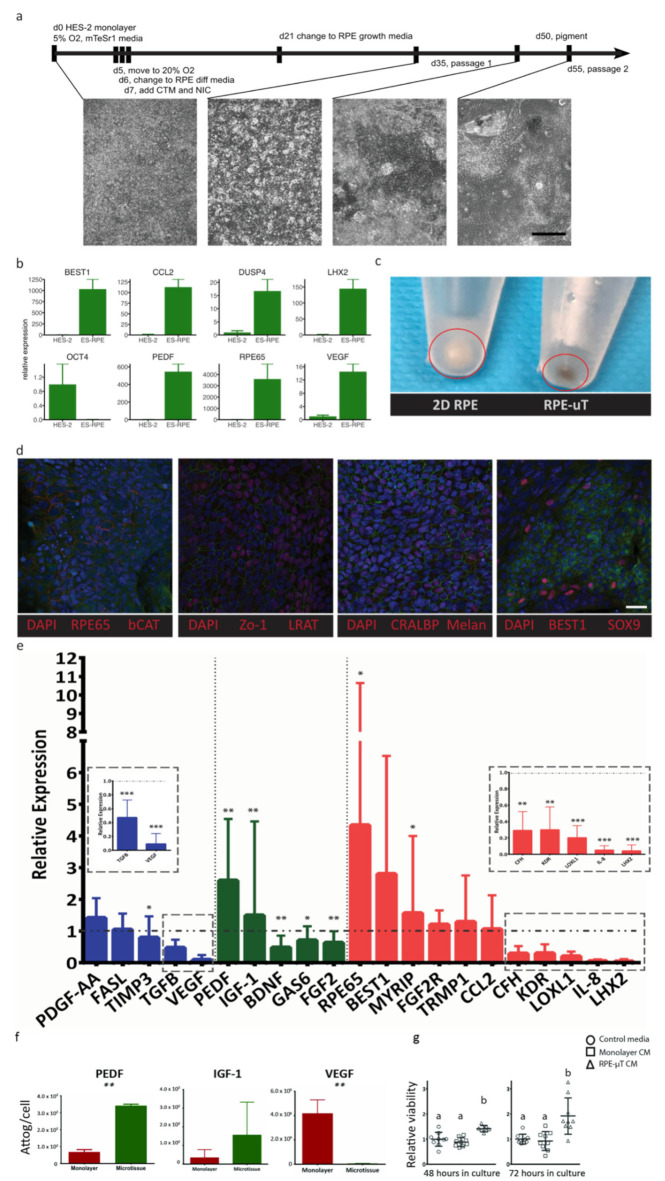
ES-derived RPE-µTs upregulate the expression of desirable photoreceptor trophic factors. (**a**) Timeline outlining culture conditions and media components at various stages of differentiation. Scale bar = 400 µm. (**b**) Relative mRNA expression levels of key RPE genes and the pluripotency marker OCT4, quantified by RT-qPCR, compared to undifferentiated HES-2 stem cells. Bars represent mean expression and error bars the maximum and minimum quantities based on three technical replicates. (**c**) ES-derived RPE-µTs appear to be more pigmented than ES-derived RPE cultured on plastic. (**d**) Moreover, newly differentiated RPE cells were immunostained for hallmark RPE markers including RPE65, CRALBP, BEST1, SOX9, Melanopsin, Zo-1 and b-CAT; cell nuclei were counterstained with DAPI. Scale bar = 25 µm. (**e**) RT-qPCR was conducted to compare the expression profile of key RPE genes between our 3D microtissue and standard adherent culture of ES-derived RPE (n = 10–11). Results were normalized to an endogenous reference gene (CNTF). The results are presented as “relative expression” values to showcase the extent of expression enhancement of certain genes in our microtissue (bar graphs) compared to adherent culture (dotted line). Genes were divided into three categories based on their function including: secreted factors for choroid stability (blue), secreted factors for photoreceptor stability (green) and other RPE functional markers (red). A Wilcoxon matched-pairs signed-ranks test was used to compare ΔCt values between RPE-µTs and adherent culture for each gene; * *p* < 0.05, ** *p* < 0.01, *** *p* < 0.001. (**f**) The amount of secreted protein in the CM, specifically PEDF, IGF-1, and VEGF, was measured by Luminex arrays (n = 5), normalized and is presented as attograms per cell. Error bars represent standard deviation. Statistical significance was determined by using a Mann–Whitney U test; ** *p* < 0.01. (**g**) Enriched mouse photoreceptors were cultured in CM from adherent RPE culture and RPE-µT. Live/dead staining was performed on cultures at 48 and 72 h after plating; mean survival ± standard deviation is shown (n = 9). Statistical significance was determined by using a Kruskal–Wallis one-way ANOVA with Dunn’s correction for multiple comparisons, groups labeled with different letters have statistically different means (*p* < 0.05).

## Data Availability

All data supporting reported results are contained within the article and the Appendix A. Raw data is available upon request to corresponding authors (D.T. and M.U.).

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
