# Peer review of "Scaffold-Free Retinal Pigment Epithelium Microtissues Exhibit Increased Release of PEDF"

_ijms, 2021, doi:10.3390/ijms222111317_

Round 1

Reviewer 1 Report

The manuscript presented is well written and the topic is novel and interesting. However, I have some minor and major concerns.

  1. In many titles of paragraph there is an extra dot at the beginning of the title (example: 2.1, 2.2 etc)
  2. Authors declare they performed in vitro assays to demonstrate the activity of their micro tissues on cell cultures. I feel these are very important data to be described and discussed.
  3. Authors claim this product is scalable. I think this should be better explained in terms of feasibility in the conclusion section. Talking about scalability, authors never address the topic of the storage of these tissues. Have they performed stability and storage tests? 

Author Response

1. In many titles of paragraph there is an extra dot at the beginning of the title (example: 2.1, 2.2 etc)

Thank you for catching this oversight. The titles have been adjusted in the following lines:

Line 123 now is “2.2 RPE-µT size influences mRNA levels of key RPE functional genes”

Line 190 now is “2.4 RPE-µT express key RPE markers”

Line 212 now is “2.5 RPE-µT upregulate transcription of desirable secreted factors”

Line 427 now is “4.3 Immunohistochemistry”

Line 441 now is “4.4 Microtissue injection modeling

Line 457 now is “4.5 Secreted protein quantification”

Line 462 now is “4.6 Viability assay”

2. Authors declare they performed in vitro assays to demonstrate the activity of their micro tissues on cell cultures. I feel these are very important data to be described and discussed.

We appreciate the reviewer’s interest in these data and have included them as Supplementary Figure 1. Our rationale for not including this data in the main figure is that we feel that this is a fairly rudimentary assay to determine the behaviour of photoreceptors in a degenerating retinal environment, although the data do suggest that biologically active trophic factors were secreted as we measured in the spent media (Figures 4 and 5). We have also expanded our discussion of these data (lines 227-33 and 326-31).

3. Authors claim this product is scalable. I think this should be better explained in terms of feasibility in the conclusion section. Talking about scalability, authors never address the topic of the storage of these tissues. Have they performed stability and storage tests? 

Thank you for your comment. We agree that scalability and storage are important considerations for determining the feasibility of using our approach. Unfortunately, we have not performed stability and storage tests. We have added a paragraph (lines 307-12) expanding on our claim of scalability and stating the necessity of assessing stability and storage of RPE-µT for clinical use.

Reviewer 2 Report

No comments

Author Response

Thank you for taking the time to review our manuscript. We are glad that you find it acceptable for publication.

Reviewer 3 Report

Comments and Suggestions for Authors

Dear authors,

The article titled ‘Scaffold-free Retinal Pigment Epithelium Microtissues Exhibit 2 Increased Release of PEDF ’ submitted by Al-Ani et al to Molecular Sciences developes new strategy in the engraftment of transplanted RPE by using a scaffold-free RPE microtissues (RPE-μT) that combine positive aspects from previous transplantation techniques.

MINOR AND MAJOR CORRECTIONS NEEDED:

ABSTRACT: Abstract is comprehensive and clear. Essential information relevant to the findings of the study is incorporated. However, PEDF in line 29 should be defined.

INTRODUCTION: The introduction is well written with enough references.

MATERIALS AND METHODS:

Please insert a paragraph about statistical analysis.

RESULTS: This section is straightforward and quite clear. The authors have explained the section based on the results obtained and categorized them. However, the following needs attention.

  1. Lines 97 – 102 might represent a good introduction of results section therefore it should precede 2.1 paragraph.
  2. Lines 140 should be “DAPI” instead of “Hoechst” Figures 3f.
  3. Figures 2d. Please add units in the scatter plot titles.
  4. Figures 2d-e. Please add R100 control or justify its absence.
  5. Figures 2e. Please correct error bars color in R400 injected group.
  6. Figures 2f. Please insert in the caption the error bars meaning (standard error or standard deviation).
  7. Figures 2f. Please provide data on the RPE-μT thickness (z-axe) in the picture, does it have a monolayer arrangement in 48h? A possible subretinal insertion of RPE-μT with diameter of 57um might represent an additional mechanical stress for a degenerated retina. Therefore, it is important to confirm a shape transiency of the engraft.
  8. Figures 2f. Please provide type of labeling.
  9. Figures 2g. Please be consistent with statistical indicators (asterisk or P value number) and error bar format.
  10. Figure 3. Please insert scale bar
  11. Figure 4a. Please use a wider format to make graph easier to read or split the graph. Please insert in the caption the error bars meaning (standard error or standard deviation). Please revise statistical analysis related to such wide error bars and possibly provide table with statistical values which can justify the statistics.
  12. Figure 4b-c-d. Please be consistent with statistical indicators (asterisk or P value number).
  13. Figure 5a. Please provide information about the scale bar.
  14. Figure 5b. Please be consistent with graph stile.
  15. Figure 5c reference is missed in the text (line 242).
  16. Figure 5d. Please insert scale bar.
  17. Figure 5e. Please use a wider format to make graph easier to read or split the graph. Please insert the error bars meaning (standard error or standard deviation) in the caption, the color legend in the graph and try to be consistent with caption: line 268 is not descriptive of y axes title. Please revise statistical analysis related to such wide error bars and possibly provide table with statistical values which can justify the statistics.
  18. Figure 5f. Please be consistent with statistical indicators (asterisk or P value number).

DISCUSSION and CONCLUSION:

  1. The authors should confirm whether RPE-μT can completely reorganize into a monolayer format.
  2. The authors produce a micro-tissue starting from epithelial cells which use to be in a monolayer format and they underline that this specific arrangement modifies gene expression pattern which gain for a possible protective effect in photoreceptors and choroid (lines 306-318). However, in lines 329-330 and in figure 2f they suggest that probably the aggregates reorganize into a monolayer quite soon. Unless the authors confirm that a disaggregated RPE-μT preserves tropic factors releasing (like PEDF, IGF-1), the therapeutic implication of the RPE-μT risks to be speculative. Accordingly showing data cited in line 305, might add weightage to the article. However, previous observation does not exclude potential RPE-μT effectiveness in replacing RPE.

Author Response

MINOR AND MAJOR CORRECTIONS NEEDED:

ABSTRACT: Abstract is comprehensive and clear. Essential information relevant to the findings of the study is incorporated. However, PEDF in line 29 should be defined.
We agree with your suggestion, and we have changed the abstract accordingly.

INTRODUCTION: The introduction is well written with enough references.MATERIALS AND METHODS:
Please insert a paragraph about statistical analysis.
Thank you for this suggestion and we apologize for the oversight. We have added a section on statistical analysis that can be found in section 4.8 (lines 485-93).

RESULTS: This section is straightforward and quite clear. The authors have explained the section based on the results obtained and categorized them. However, the following needs attention.
1.    Lines 97 – 102 might represent a good introduction of results section therefore it should precede 2.1 paragraph.
Thank you for this suggestion! We agree that the referred to paragraph would serve as a good introduction to the results section. As such, we made the necessary changes (line 96-101).

2.    Lines 140 should be “DAPI” instead of “Hoechst” Figures 3f.
Thank you for catching this oversight on our part. This has been corrected.

3.    Figures 2d. Please add units in the scatter plot titles.
The units (arbitrary relative fluorescence units, ‘RFU’) were added to both axes. 

4.    Figures 2d-e. Please add R100 control or justify its absence.
Due to a technical error that we did not catch in time, the green channel for these images was over-exposed. As a result, over 80% of the pixels were at full saturation and these images could not be quantified to add to Figures 2 d-e. Due to limited lab access we are not able to repeat this experimental series. We have made note of this in the figure legend and added an explanation to the text (lines 143-7) to justify its absence.

5.    Figures 2e. Please correct error bars color in R400 injected group.
Thank you for pointing that out. The figure has been corrected as per your suggestion. 

6.    Figures 2f. Please insert in the caption the error bars meaning (standard error or standard deviation).
Error bars represent the standard deviation throughout the manuscript. We have adjusted all figure legends to reflect this and apologize for the omission. 

7.    Figures 2f. Please provide data on the RPE-μT thickness (z-axe) in the picture, does it have a monolayer arrangement in 48h? A possible subretinal insertion of RPE-μT with diameter of 57um might represent an additional mechanical stress for a degenerated retina. Therefore, it is important to confirm a shape transiency of the engraft.

We believe that our aggregates do adopt a monolayer arrangement and looked at confocal image stacks to determine that over 48 h the aggregate height decreases considerably, from 57 µm to less than 30 µm and have included this in the main text (158-60). While there is potential for additional mechanical stress when injecting large RPE-µT, we feel that this will likely be minimal as it is considerably less than the focal length of the eye (lines 366-9). This further emphasizes the importance of in vivo experiments to confirm the benefit of RPE-µT in the subretinal space.

8.    Figures 2f. Please provide type of labeling.
Thank you for catching this oversight. The legend for figure 2f has now been updated to include mCherry as a nuclear marker for these cells. All cells used in this experiment ubiquitously express mCherry as verified by flow cytometry.

9.    Figures 2g. Please be consistent with statistical indicators (asterisk or P value number) and error bar format.
Adjusted as per your suggestion. 

10.    Figure 3. Please insert scale bar.
Thank you for catching this oversight. We have added a 20 μm scale bar on figure 3h composite image. 

11.    Figure 4a. Please use a wider format to make graph easier to read or split the graph. Please insert in the caption the error bars meaning (standard error or standard deviation). Please revise statistical analysis related to such wide error bars and possibly provide table with statistical values which can justify the statistics.
Thank you for your insightful suggestions. Although we maintained the current format of the figure, as we believe it provides a side-by-side comparisons among the various genes, we have changed the size of the various panels and increased the width of the figure. We hope that this change improves readability. 
We have included in all figure legends the meaning of the error bars. While our data does have high variability, we chose to perform a one-sample Wilcoxon signed rank test to test whether the mean expression in RPE-µT was significantly different from a value of 1, the reference monoculture sample. We have made this explicit in both figure legends (Figures 4 and 5) and our description of the statistical analysis performed (Section 4.8).

12.    Figure 4b-c-d. Please be consistent with statistical indicators (asterisk or P value number).
Thank you for flagging these sections as well. The figure has been adjusted according to your suggestion. 

13.    Figure 5a. Please provide information about the scale bar.
This has been added.

14.    Figure 5b. Please be consistent with graph stile.
We updated this graph to match the style of the other graphs in our manuscript.

15.    Figure 5c reference is missed in the text (line 242).
Thank you for pointing that out. The approperiate in-text citation has been added (lines 255-7).

16.    Figure 5d. Please insert scale bar.
This has been added.

17.    Figure 5e. Please use a wider format to make graph easier to read or split the graph. Please insert the error bars meaning (standard error or standard deviation) in the caption, the color legend in the graph and try to be consistent with caption: line 268 is not descriptive of y axes title. Please revise statistical analysis related to such wide error bars and possibly provide table with statistical values which can justify the statistics.

To address your thoughtful comments, we have revised Figure 5e as per your suggestions for Figure 4a, above: 
a)    We have adjusted figure 5e to a wider format that is easier to read. 
b)    We have we have stated that error bar represent standard deviation.
c)    We adjusted the caption to be consistent with the y-axis title ‘relative expression’.
d)    We explicitly state the statistical test used both in the figure legend and section 4.8.

18.    Figure 5f. Please be consistent with statistical indicators (asterisk or P value number).
Thank you for pointing that out. We changed the statistical indicators to asterisks to be consistent with our other figures.

DISCUSSION and CONCLUSION:
19.    The authors should confirm whether RPE-μT can completely reorganize into a monolayer format.

Our study proposes that RPE-μT have better therapeutic value due to their improved genetic profile when compared to a conventional monolayer of RPE cells. We further attempted to illustrate that RPE-μT can spread and form a monolayer that is compatible with the in vivo RPE environment since having a sphere of cells (RPE-μT) would not satisfy physiological needs. While we were able to include the approximate height of aggregates after plating on plastic culture-ware, we unfortunately are unable to conduct additional experiments to confirm that a true monolayer (e.g., the establishment of tight junctions) was formed although this provides an excellent stepping off point for future studies.

20.    The authors produce a micro-tissue starting from epithelial cells which use to be in a monolayer format and they underline that this specific arrangement modifies gene expression pattern which gain for a possible protective effect in photoreceptors and choroid (lines 306-318). However, in lines 329-330 and in figure 2f they suggest that probably the aggregates reorganize into a monolayer quite soon. Unless the authors confirm that a disaggregated RPE-μT preserves tropic factors releasing (like PEDF, IGF-1), the therapeutic implication of the RPE-μT risks to be speculative. Accordingly showing data cited in line 305, might add weightage to the article. However, previous observation does not exclude potential RPE-μT effectiveness in replacing RPE.
Thank you for this comment and kind summary of our work. The main purpose of generating RPE-μT was to circumvent the current challenges encountered with single-cell suspension and RPE sheets (lines 66-73 and 84-9). The value of RPE-μT is further supported by their ability to reorganize into a monolayer on a Bruch’s membrane analogous substrate, Matrigel. Your comment hints at exciting future investigations comparing the initial monolayer that was used to manufacture the RPE-μT with RPE-μT that has been allowed to reorganize into a monolayer, which is unfortunately beyond the scope of our current work. We suspect that since the cells are not mechanically dissociated, much of their improved transcriptome will remain although more research is needed. We further expanded our discussion of the in vitro photoreceptor assay (lines 227-33 and 326-31) and included this data as Supplementary Figure 1 to further support the advantages of RPE-μT over monolayer culture.

Round 2

Reviewer 1 Report

I thank the authors for their work and their answers. I still believe that the in vitro test authors reported in Supplementary F1 should be included and commented in the main text, even if preliminary.

Nevertheless, I recommend publication of this work.

Author Response

Thank you for your kind comments. We agree with your suggestion and have moved the in vitro test data to the main text (Fig. 5g).

Reviewer 3 Report

I thank the authors for all their corrections. I personally think that including supplementary figure 1 in the manuscript can provide valuable weightage to the study. For the same reason, it could be useful to insert a table about original values from RTq-PCR analysis as supplementary data. 

Author Response

Thank you for your helpful comments, which have strengthened our manuscript. We agree with your suggestions and have included the in vitro photoreceptor data as part of the main manuscript (Fig. 5g) and added supplementary tables with the summary statistics for our RT-qPCR experiments.